# “Safer Births Bundle of Care” Implementation and Perinatal Impact at 30 Hospitals in Tanzania—Halfway Evaluation

**DOI:** 10.3390/children10020255

**Published:** 2023-01-30

**Authors:** Hege Ersdal, Paschal Mdoe, Estomih Mduma, Robert Moshiro, Godfrey Guga, Jan Terje Kvaløy, Felix Bundala, Boniphace Marwa, Benjamin Kamala

**Affiliations:** 1Department of Anesthesia, Stavanger University Hospital, 4011 Stavanger, Norway; 2Faculty of Health Sciences, University of Stavanger, 4021 Stavanger, Norway; 3Haydom Lutheran Hospital, Haydom 9000, Tanzania; 4School of Public Health and Social Sciences, Muhimbili University of Health and Allied Sciences, Dar es Salaam 65001, Tanzania; 5Department of Pediatrics, Muhimbili National Hospital, Dar es Salaam 65000, Tanzania; 6Department of Mathematics and Physics, University of Stavanger, 4021 Stavanger, Norway; 7Department of Research, Stavanger University Hospital, 4011 Stavanger, Norway; 8Reproductive and Child Health Section, Ministry of Health, Dodoma 743, Tanzania; 9Department of Health, President’s Office Regional Authority and Local Government, Dodoma 1923, Tanzania

**Keywords:** newborn resuscitation, simulation-based training, safer births, quality improvement, perinatal mortality, newborn mortality, fresh stillbirths, maternal mortality, helping babies breathe, helping mothers survive

## Abstract

Safer Births Bundle of Care (SBBC) consists of innovative clinical and training tools for improved labour care and newborn resuscitation, integrated with new strategies for continuous quality improvement. After implementation, we hypothesised a reduction in 24-h newborn deaths, fresh stillbirths, and maternal deaths by 50%, 20%, and 10%, respectively. This is a 3-year stepped-wedged cluster randomised implementation study, including 30 facilities within five regions in Tanzania. Data collectors at each facility enter labour and newborn care indicators, patient characteristics and outcomes. This halfway evaluation reports data from March 2021 through July 2022. In total, 138,357 deliveries were recorded; 67,690 pre- and 70,667 post-implementations of SBBC. There were steady trends of increased 24-h newborn and maternal survival in four regions after SBBC initiation. In the first region, with 13 months of implementation (*n* = 15,658 deliveries), an estimated additional 100 newborns and 20 women were saved. Reported fresh stillbirths seemed to fluctuate across time, and increased in three regions after the start of SBBC. Uptake of the bundle varied between regions. This SBBC halfway evaluation indicates steady reductions in 24-h newborn and maternal mortality, in line with our hypotheses, in four of five regions. Enhanced focus on uptake of the bundle and the quality improvement component is necessary to fully reach the SBBC impact potential as we move forward.

## 1. Introduction

Deaths related to childbirth, including young women, unborn and newborn babies, are still a huge global concern and challenge. Approximately 98% of these perinatal deaths occur in low- and middle-income countries, with 50% in sub-Saharan Africa [1]. Overall global maternal mortality is estimated to be around 211 deaths per 100,000 live births [2], but in Tanzania, a sub-Saharan country, as many as 556 deaths per 100,000 births are reported [3]. This means that 1 in 33 women die in relation to pregnancy and childbirth, with postpartum haemorrhage as the leading cause of death. The global estimates for stillbirths and newborn mortality are 13.9 and 17.6 per 1000 births, respectively [4,5]. In Tanzania, the burden is higher; there are 39 stillbirths per 1000 births and 25 newborn deaths per 1000 live births [3]. The majority of these deaths are intricately linked to obstetric complications, and are often related to sub-optimal healthcare around the time of labour and birth [1,3,4,5,6]. Thus, perinatal deaths (i.e., intrapartum related maternal deaths, fresh stillbirths (FSB), and newborn deaths) can be substantially reduced by improved quality of care around the time of birth [7].

The “Safer Births Bundle of Care” (SBBC) package was co-developed in Tanzania to help health care workers (HCW) improve the quality of labour and newborn care [8]. SBBC consists of a bundle of proven innovative clinical [9,10,11,12,13,14] and training tools [15,16,17], combined with new strategies for instituting continuous quality improvement (CQI) efforts and sustainable improvements in care [18,19,20]. Key components of the CQI efforts are (1) regular, on-the-job, low-dose, high-frequency simulation-based training; and (2) utilisation of local data and feedback loops to visualise gaps in clinical care and guide ongoing training needs. Adequate training of local champions who can facilitate CQI and simulation training is considered essential for these processes to happen, and to stimulate a gradual and sustainable culture change [9]. SBBC is a collaboration among several Tanzanian and Norwegian partners, and has the potential for global scale-up if proven efficient and effective in saving lives. In 2019, SBBC was awarded an innovation-to-scale fund from the Global Financing Facility, an arm of the World Bank, to test implementation of the SBBC package in 30 healthcare facilities across five regions in mainland Tanzania. The roll-out is led by Haydom Lutheran Hospital, in close collaboration with the Ministries of Health and UNICEF in Tanzania.

The aim of this study is to describe pre-implementation baseline data and halfway post-implementation trends across all sites in (1) adoption of clinical tools, (2) key clinical performance indicators, and (3) perinatal deaths (i.e., 24-h newborn mortality, FSB, and maternal mortality).

## 2. Materials and Methods

### 2.1. Organisation, Study Sites, and Population

Haydom Lutheran Hospital, in collaboration with the two Ministries of Health in Tanzania (National Ministry of Health and President’s Office, Regional Authority and Local Government), professional bodies (Pediatric Association of Tanzania and Tanzania Midwives Association), SAFER, Stavanger University Hospital, Laerdal Global Health (Norway), and UNICEF Tanzania, is responsible for implementing SBBC in five selected regions in Tanzania: Manyara, Tabora, Geita, Shinyanga, and Mwanza. Each region represents a wedge consisting of six health facilities (clusters). The five regions and 30 facilities were selected in collaboration with the Ministries of Health. The inclusion criteria for regions/facilities were a high burden of perinatal mortality, a high volume of deliveries, and no other relevant interventions. All the facilities are categorised as Comprehensive Emergency Obstetric and Newborn Care (CEmONC). They represent different levels of the health system: regional/referral hospitals (one in each region), district hospitals, and health centres. The annual number of births ranges from 1000–10,000 across the sites. All parturient women with a suspected live fetus at the start of labour and their offspring (>28 weeks gestation) are enrolled in the study.

### 2.2. Study Design and Timelines

The roll-out follows a stepped-wedge cluster randomised implementation design over a 3-year period, from March 2021 to December 2023 [9] (ISRCTN Registry: ISRCTN30541755). The first region, Manyara, was purposively selected for logistical and strategic reasons, whereas subsequent regions were selected randomly, using simple randomisation. An overview of regions, facilities, and time periods for pre-implementation (baseline) and post-implementation data collection is presented in Figure 1. Due to early trends of lower mortality in the first regions, increased understanding of implementation requirements among the SBBC team, and requests from the regional health authorities, Mwanza (the last region) was allowed to begin implementation around the same time as Shinyanga, two months earlier than scheduled.

### 2.3. Safer Births Bundle of Care Components and Interventions

SBBC is the result of 10 years of multidisciplinary innovation and research, mainly undertaken in Tanzania and Norway. The bundle has four main components: training innovations, clinical innovations, systematic CQI interventions, and creating the infrastructure and competence for sustainability. In combination, these components shall mutually strengthen each other and support a sustainable improvement in care and patient outcomes.

The SBBC package was co-created in Tanzania to facilitate clinical care and CQI through frequent on-the-job simulation training (Figure 2). The innovative clinical tools (Laerdal Global Health, Stavanger, Norway) are the following: Moyo Fetal Heart Rate Monitor for improved labour management and prevention of FSB and severe birth asphyxia [10,11,12]; NeoBeat Newborn Heart Rate Meter for immediate assessment at birth (e.g., distinguish true FSB from newborns in urgent need of resuscitation) and guiding resuscitation attempts [13]; and Upright Newborn Bag and Mask for improved bag-mask ventilation of non-breathing newborns [14]. The innovative training tools (Laerdal Global Health, Stavanger, Norway) are the following: NeoNatalie Live Newborn Ventilation Trainer, a simulator for newborn resuscitation training [15,16]; and MamaNatalie Postpartum Hemorrhage Simulator for training in labour management and identification, prevention, and management of postpartum bleeding [17]. The systematic CQI interventions are supposed to be driven by trained 2–3 local “facility champions” based at each facility [18,19]. These facility champions were trained by national facilitators, supported by SAFER simulation experts, before beginning SBBC implementation in each region. More details about the training courses and cascade have been published previously [9]. The role of the local champions is to facilitate regular simulation training on labour management, postpartum bleeding, newborn resuscitation, and essential newborn care. The training sessions are guided by the Helping Mothers and Babies Survive programs developed by Jhpiego and the American Academy of Pediatrics with partners [21,22]. The local champions receive weekly feedback on their own facility’s clinical data (key performance indicators and perinatal outcomes) and should adjust ongoing training to address identified gaps and challenges.

Prior to the start of baseline data collection, sensitisation meetings were held in November 2020. Key local, regional, and national stakeholders/authorities, together with hospital management teams from all regions, participated (*n* = 250). These meetings were arranged to inform, create awareness, and establish ownership of SBBC.

At the start of SBBC implementation, each facility received the full package of innovative tools and training. Since the initiation of implementation, scheduled supportive supervisions and mentorship visits continue to take place at all regions. These are led by the trained national facilitators, in collaboration with the Tanzanian SBBC investigators and regional coordinators [9].

Clinical data are collected semi-automatically by the new devices Moyo, NeoBeat, Upright, through the Liveborn App system. Weekly presentations of local clinical data (performance and patient outcomes) and training data are available for rapid quality improvement loops and targeted training. Newborn training data are collected semi-automatically by the NeoNatalie Live system. During postpartum bleeding training using the MamaNatalie simulator, data are recorded in logbooks.

### 2.4. Data Collection and Management

In December 2020, prior to the introduction of SBBC, systematic readiness assessments were conducted at each site. The aim was to evaluate strengths and weaknesses, and identify bottlenecks related to the readiness, availability, and quality of provided intrapartum care. These preliminary findings were used to tailor implementation of the bundle at each health facility.

In January 2021, 60 data collectors (2 for each site) and 5 regional coordinators (1 for each region) were trained to conduct and oversee data collection. They manage daily quality control procedures, including correction of errors and safe transfer of data to the central research server at Haydom.

Prospective baseline data collection at all sites began 1 March 2021. Relevant indicators for labour and newborn care, as well as maternal and newborn characteristics and outcomes, are gathered from partograms and patient files. These data points are entered into an electronic data collection system (i.e., ODK) by the data collectors, and are uploaded to the central server every day. A data manager at Haydom performs a final quality control check before storing the data. Any identified errors/queries must be resolved by the regional coordinator and responsible data collectors. The central data manager summarises basic statistics with selected key indicators that are reported back to each facility on a weekly basis, in order to serve as a basis for local CQI activities (Figure 2).

### 2.5. Statistical Methods

For this halfway evaluation, key clinical performance variables, maternal and newborn characteristics, and outcomes are described across the five regions, before and after SBBC implementation. Categorical data are presented as percentages (numbers), and continuous data as medians (quartiles). To test for differences between regions, chi-square tests are used for categorical data and Kruskal–Wallis tests are used for continuous data. *p* < 0.05 is used as level of significance.

Monthly proportions are plotted to illustrate adoption of Moyo and NeoBeat over time and changes in patient mortality rates. To further illustrate and quantify changes in patient survival, variable life-adjusted display (VLAD) plots are presented, demonstrating the estimated cumulative number of lives saved over time, compared to the baseline situation [23]. Data are analysed using R-4.2.1 open-source software [24].

### 2.6. Ethical Considerations

SBBC was approved by the National Institute for Medical Research (Ref. NIMR/HQ/R.8a/Vol.IX/3458) in Tanzania, and the Regional Ethical Committee in Norway (Ref. 229725). Approvals were also obtained from all facility managements. All women admitted to the labour ward for delivery are informed about the quality improvement project and the new clinical tools (Moyo, NeoBeat, and Upright). Consent was not deemed necessary for this quality improvement project. All data are de-identified. Permission to publish this paper was obtained from the National Institute for Medical Research, Tanzania (Ref. No: NIMR/HQ/P.12 VOL XXXV/90).

## 3. Results

### 3.1. Patient Characteristics and Key Performance Indicators

In total, 138,357 deliveries occurred between 1 March 2021 and 31 July 31 2022; 67,690 deliveries were recorded in the pre-implementation period, and 70,667 were recorded in the post-implementation period. Baseline data, including maternal and newborn characteristics, key performance indicators, and patient outcomes for the five regions, are presented in Table 1. The same variables post-implementation are presented in Table 2. For all variables, the observed differences between regions are statistically significant (*p* < 0.0001), likely because of the large number of births. We consider some of the differences to be of clinical importance, both during baseline and after initiation of SBBC. The clinically important differences are related to numbers of referrals (i.e., women in labour admitted from other facilities), fetal heart rate measurements, numbers of newborns being bag-mask ventilated, and numbers of reported FSB. Tabora seems to have more referrals than the other regions, higher numbers of babies with non-detectable fetal heart rate on admission and during labour, more newborns receiving ventilation, and more FSB. Across regions, fetal heart rate is not recorded in 10–30% of deliveries, both before and after SBBC (Table 1 and Table 2). Before start of SBBC, very few facilities had access to Moyo and NeoBeat. After introduction of SBBC, all facilities had enough devices to use Moyo and NeoBeat on most deliveries/newborns. The adoption/uptake rate of use varies across regions, from 20–66% of deliveries monitored with Moyo, and 2.9–9.6% of all newborns monitored with NeoBeat (Table 2). Figure 3 illustrates the uptake of Moyo among all births for each region (a–e) over time. Figure 4 illustrates the uptake of NeoBeat among newborns being stimulated and/or ventilated for each region (a-e) over time.

### 3.2. Perinatal Outcomes and Use of Moyo and NeoBeat

Numbers of reported 24-h newborn and maternal deaths seem to be decreasing in four regions (Manyara, Geita, Shinyanga, and Mwanza) since the start of SBBC. The number of reported FSB are almost similar (Manyara, Shinyanga) or worse (Tabora, Geita, Mwanza) post-implementation. Absolute numbers are presented in Table 1 and Table 2. Figure 5, Figure 6 and Figure 7 illustrate mortality rates and estimated numbers of additional lives saved after introduction of SBBC for 24-h newborn deaths (Figure 5), FSB (Figure 6), and women (Figure 7) for all regions (a–e) over time. In Manyara, with 4936 deliveries in baseline and 15,658 deliveries after intervention start in July 2021, an estimated additional 100 newborns and 20 women have been saved over the 13-month implementation period (Figure 5a and Figure 7a).

Table 3 presents baseline and after SBBC data for FSB, dead and alive newborns in Manyara and Tabora, the two regions with longest post-implementation periods. During the baseline period in both regions, more FSB and 24-h newborn deaths occurred after referral/admission from other facilities or the antenatal care/ward during labour, compared to newborns still alive at 24 h. Irrespective of the admission route, FSB and those who died had a non-detectable or not measured/recorded fetal heart rate on admission more frequently than among those who were still alive at 24 h. After initiation of SBBC in Manyara, 9.6% of newborns who died were admitted during labour from the antenatal care/ward. NeoBeat was used in 69.2% of those who died, and bag-mask ventilation was provided in 78.8% compared to 34.8% before SBBC training. Among those eventually classified as FSB, more seemed to be referred from other facilities after start of SBBC (11.8% versus 3.3%); NeoBeat was used in 6.5% and bag-mask ventilation was attempted in 6.5% of FSB. After initiation of SBBC in Tabora, 17.6% of newborns who died, and 41.3% of FSB, were referred during labour from other facilities. NeoBeat was used in 72.1% of those who died, and bag-mask ventilation was provided in 83.8% compared to 63.1% before SBBC training. Among those eventually classified as FSB, NeoBeat was used in 6.2% and bag-mask ventilation was attempted in 9.4%.

### 3.3. Maternal Mortality

During baseline in Manyara, none of the postpartum maternal deaths were admitted in labour from other facilities. After initiation of SBBC, 75% of maternal deaths were admitted in labour from other facilities or antenatal care/ward, and 75% were secondary to postpartum bleeding. In Tabora, 30% and 42% of the maternal deaths were admitted in labour from other facilities or antenatal care/ward during baseline and after start of SBBC, respectively.

## 4. Discussion

In this halfway report of SBBC implementation in five Tanzanian regions, we present encouraging and steady trends of increased 24-h newborn survival and maternal survival in four of five regions. In these four regions, our original hypotheses of a 50% reduction in 24-h newborn deaths and 10% reduction in maternal deaths, seem to be within reach. For FSB, the trends fluctuate more, and in three of five regions an increased number of FSB have been reported after introduction of SBBC. Maternal and newborn characteristics are quite similar across all five regions, both pre- and post-implementation. However, some key performance indicators and adoption rates of the new devices vary across facilities, and these differences may influence mortality rates.

The current SBBC roll-out follows a stepped-wedge cluster randomised design, starting in Manyara region, followed by Tabora. At the time of this halfway evaluation (end of July 2022), Manyara had been implementing SBBC over 13 months and Tabora over 11 months. Therefore, the most reliable preliminary trend data can be obtained from these two regions. Descriptive numbers from these two regions indicate large differences in the use of the new devices, clinical performance indicators, and patient outcomes. There are important differences in variables representing pathways to FSB and 24-h newborn deaths, such as being referred during labour from another facility, having abnormal, non-detectable or not measured fetal heart rate, and/or receiving ventilation at birth [25,26,27]. In Tabora, which reports worse perinatal outcomes than Manyara, as much as 42% of the 276 reported FSB were referrals from other facilities, and more than half of the FSB had a non-detectable or abnormal fetal heart rate on admission. On the contrary, in Manyara, with fewer overall referrals, around 12% of the 93 reported FSB were admitted from other facilities, and 4.5% from antenatal care/wards. The latter indicates that women experiencing complicated labour are transferred too late from “waiting compounds” within the same facility. More timely transfer to the labour ward (within the same facility) will be included as a CQI effort during the final phase of this project (Figure 1).

The stable and sustained trend of increased 24-h newborn survival in Manyara indicates that the SBBC package enables HCW to save more lives, with its innovative tools for better clinical care, more efficient training, and local establishment of CQI efforts. Surprisingly, the proportion of newborns being ventilated seems to be decreasing slightly from baseline (4.2%) to after implementation (3.9%). We expected an increase in newborns being ventilated after introduction of SBBC, due to the relatively high perinatal mortality rates and reported figures from a similar setting which documented that around 8% of newborns needed ventilation [19]. Therefore, we fear that several newborns who would benefit from this life-saving intervention still do not receive it. A strengthened focus on CQI loops, linking newborn treatment and outcome measures with targeted resuscitation training at each site, may be necessary to close this potential gap (Figure 2).

Furthermore, we believe newborns in both Manyara and Tabora would benefit from more extensive use of NeoBeat to help determine true FSB and identify newborns in urgent need of ventilation [13]. The overall uptake of NeoBeat use is not very high, but twice as high in Manyara (9.6% of all births) as in Tabora (4.6% of all births). Furthermore, NeoBeat was only used on around 6% of those classified as FSB in both regions, even if NeoBeat can help distinguish true FSB from severely asphyxiated newborns [25].

The uptake of Moyo use is almost double in Manyara compared to Tabora, fluctuating at around 66% and 36% of all deliveries, respectively. However, we note that the proportion of deliveries with not measured/recorded fetal heart rate is decreasing in Tabora, from around 27% before the start of SBBC, to around 16% since implementation. Despite the improvements, it is a remaining challenge that too many deliveries are not followed up with adequate fetal heart rate monitoring, both in Manyara and Tabora, representing a missed opportunity to prevent both intrapartum deaths, i.e., FSB, and severely asphyxiated newborns with a high risk of dying [25,26,27]. Even if there is a relatively good uptake of Moyo use in Manyara, fetal heart rate was not measured and/or recorded at all during labour in 23% of all deliveries. This continued high number may be one important reason for the lack of improvement in FSB trends, in addition to those being admitted after prolonged labour in lower-level facilities (without access to SBBC interventions). Adequate fetal heart rate monitoring is the first essential step to identify babies at risk during labour.

The maternal mortality trend in Manyara is better than expected, showing an almost 20% reduction after the start of SBBC. The same encouraging early trend is also being seen in Geita, Shinyanga, and Mwanza. At this stage, we can only report that there has been a great focus on implementing good Helping Mothers Survive Bleeding After Birth simulation training during roll-out in all he 30 facilities [17,28].

As stated above, Manyara and Tabora have been presented and discussed in more detail, since these are the regions with the longest implementation period. The remaining regions were between 5–8 months into SBBC implementation at the time of this halfway evaluation. Importantly, the early trends in the three last regions are quite similar to Manyara, showing a steady increase in 24-h newborn survival and maternal survival, but a reported increase in FSB. Although the time periods for adoption, training, and changing behaviour are shorter in the last regions, the early impact trends may be observable due to the high numbers of deliveries. Furthermore, the reported mortality rates, both before and after start of SBBC, are quite similar in Manyara, Geita, Shinyanga, and Mwanza, whereas Tabora is standing out with worse mortality rates both prior to and after introduction of SBBC (Table 1 and Table 2). These variations may indicate different facility readiness levels at the start of SBBC.

The use/uptake of Moyo and NeoBeat also varies across the regions, from 20–66% of all deliveries being monitored with Moyo, and 2.9–9.6% of all newborns being monitored with NeoBeat after start of SBBC. These numbers are lower than expected, since all sites are provided with enough devices to use Moyo on most deliveries (Figure 3) and NeoBeat on all newborns who are not vigorous and/or crying immediately after birth (Figure 4).

Several SBBC studies are underway to investigate further the documented differences across regions, reasons for improvements or no improvements, and why some sites seem to adopt and change behaviour faster than others. We plan to use the scoring reports from the readiness assessment analyses and the mentoring/supervision follow-ups of each site to assess associations between facility readiness level and the ability to implement a new complex intervention such as SBBC. The SBBC package is extensive, and we want to better understand limitations in integrating and adopting several interventions/changes at the same time. Furthermore, we will study the SBBC training interventions and potential impacts on clinical changes, and how different sites are able to utilise the CQI component with local data-driven feedback loops to guide ongoing training and other QI efforts—addressing gaps in clinical care. We believe strengthening of local data collection, reporting, feedback, and use to be fundamental for institutionalisation of CQI loops, and to achieve maximal impact of SBBC. Following data collection over time in all regions, we see a clear trend of fewer missing values after the start of SBBC (manuscript in preparation), and speculate that the constructive CQI program contributes to a decreased “shame and blame” culture, an increased understanding of the usefulness of accurate data, and a potential for more registration of adverse outcomes over time [28,29,30,31,32]. This may, to some extent, explain the unexpected increase in reported FSB after the start of the intervention. Furthermore, we believe that a continued focus on building competence and capacity to run local simulation-based training and CQI efforts based on own accurate data, are necessary to create sustainability.

The main limitations of our study are the following: (1) suspected inaccurate data reporting/collection and underreporting of adverse outcomes, especially in the baseline period; (2) a potential contamination risk of the baseline and/or control periods of later wedges/clusters, which may dilute the effects; (3) a slight disruption of the randomisation sequence, as the last region began SBBC implementation two months before schedule; and finally, (4) difficulty in using the CONSORT diagram to report our project, as a number of clusters (facilities) have been allocated to a number of wedges (regions), and the crossovers from baseline to implementation take place at different times.

## 5. Conclusions

In conclusion, this halfway evaluation of SBBC indicates steady trends toward a 50% reduction in 24-h newborn mortality, and a 10–20% reduction in maternal mortality in four of five regions. Reported FSB rates seem to be stable in two regions and worse in three regions. We described differences between the regions and potential relations between key performance clinical indicators and perinatal mortality, and hence, areas for improvements as we move forward. Despite increased survival rates that are in line with our hypotheses, these preliminary data indicate that there is a remaining impact potential if uptake of the bundle is further strengthened over time.

## Figures and Tables

**Figure 1 children-10-00255-f001:**
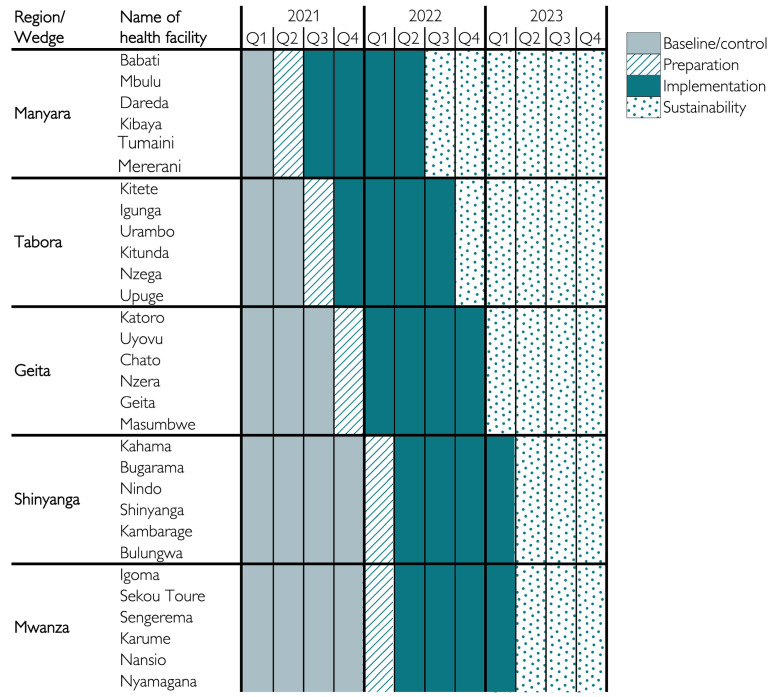
Time periods for interventions in the different regions (wedges) and health facilities (clusters) during roll-out of SBBC.

**Figure 2 children-10-00255-f002:**
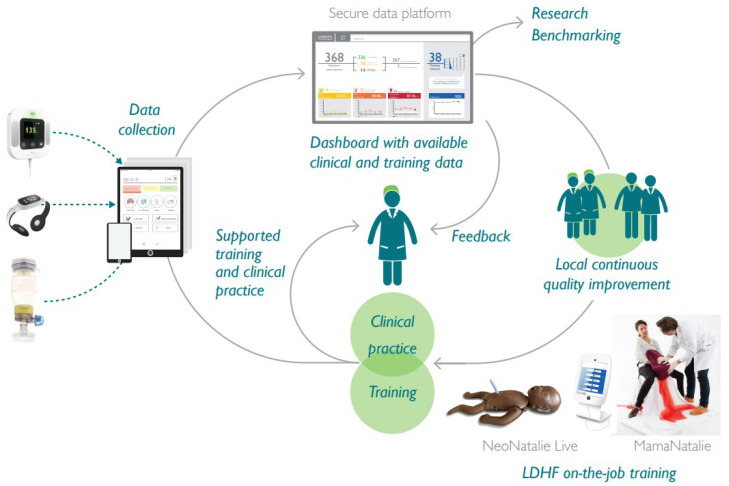
Integration of the innovative SBBC tools for clinical care, data collection, and training to facilitate continuous quality improvement efforts. LDHF = low-dose high-frequency. Photo is reprinted with permission from Laerdal Global Health.

**Figure 3 children-10-00255-f003:**
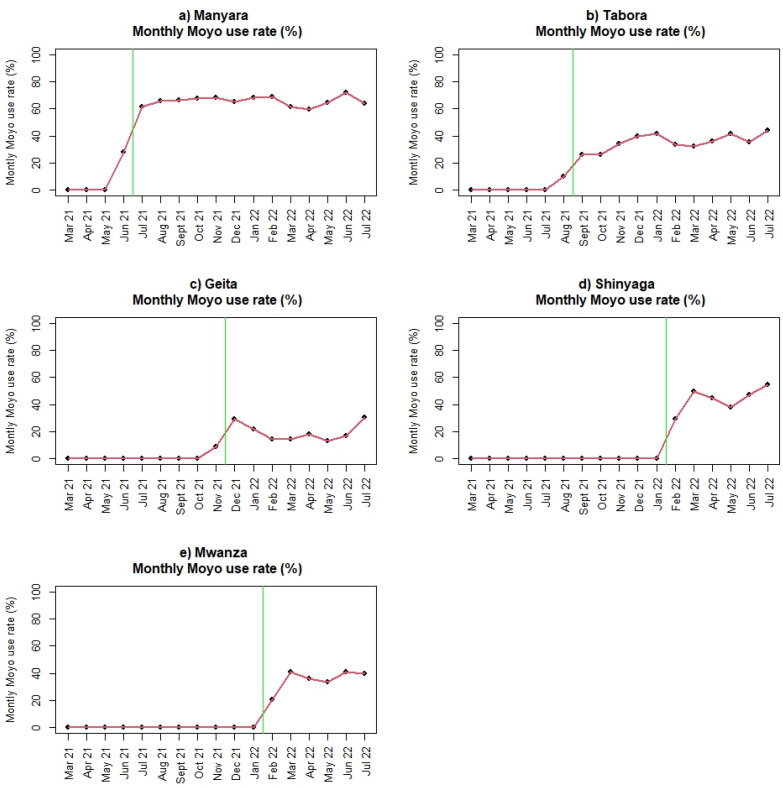
(**a**–**e**): Use of Moyo among all deliveries before and after introduction of SBBC (green line) for (**a**) Manyara, (**b**) Tabora, (**c**) Geita, (**d**) Shinyanga, and (**e**) Mwanza regions.

**Figure 4 children-10-00255-f004:**
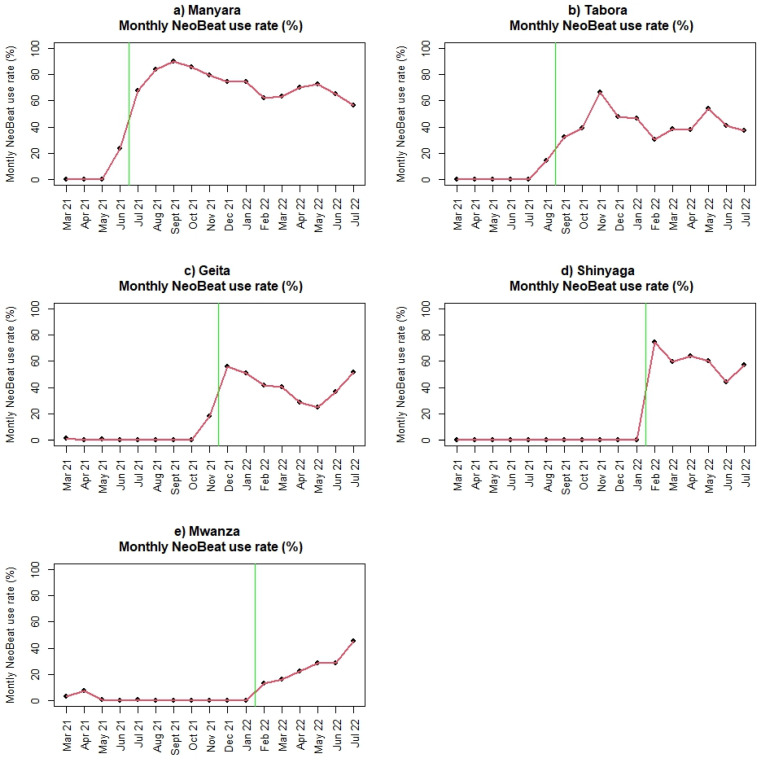
(**a**–**e**): Use of NeoBeat among newborns being stimulated and/or ventilated before and after introduction of SBBC (green line) for (**a**) Manyara, (**b**) Tabora, (**c**) Geita, (**d**) Shinyanga, and (**e**) Mwanza regions.

**Figure 5 children-10-00255-f005:**
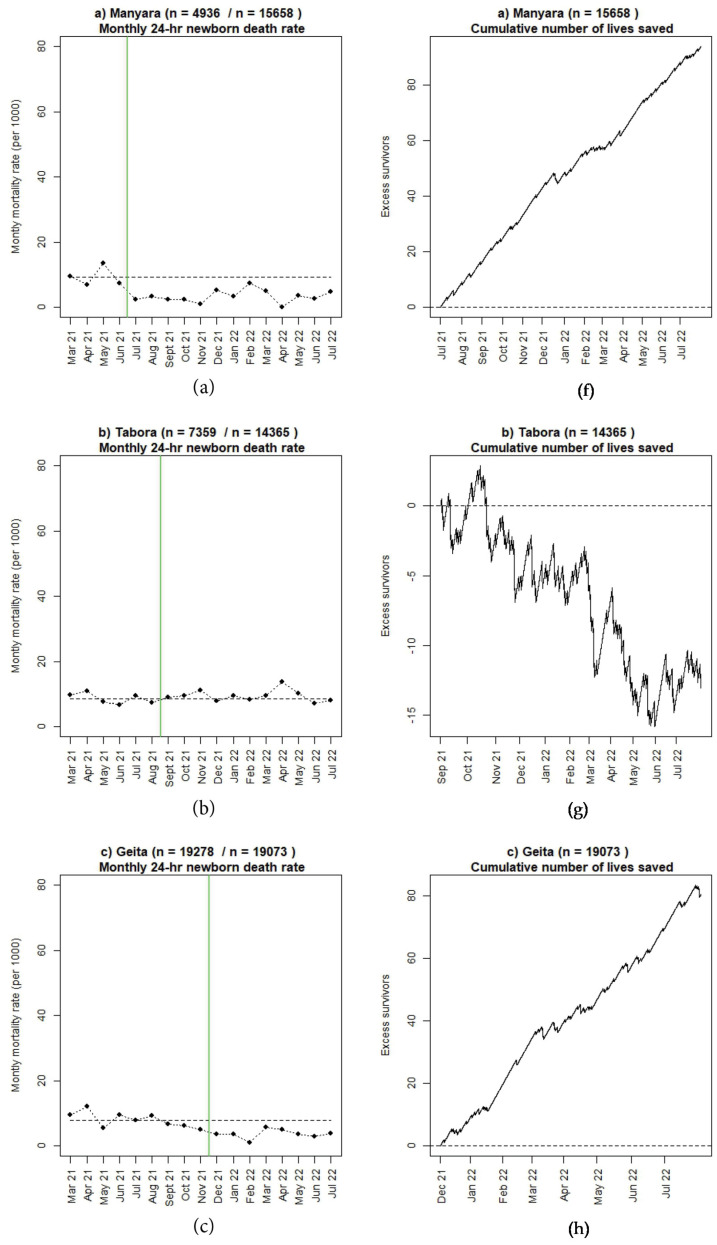
(**a**–**e**): Monthly 24-h newborn mortality rate before and after introduction of SBBC (green line) and estimated cumulative numbers of additional lives saved after introduction of SBBC for (**f**) Manyara, (**g**) Tabora, (**h**) Geita, (**i**) Shinyanga, and (**j**) Mwanza regions.

**Figure 6 children-10-00255-f006:**
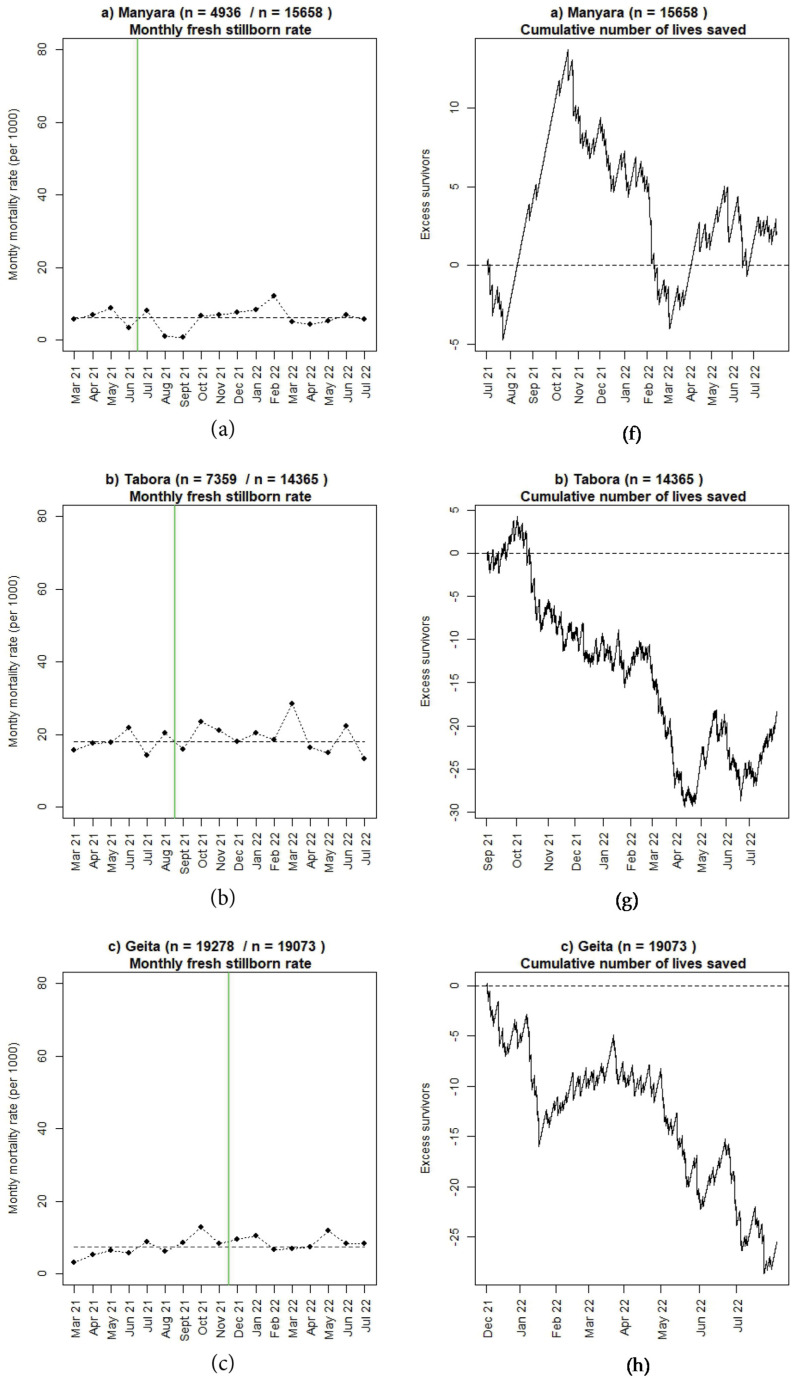
(**a**–**e**): Monthly fresh stillbirth rate before and after introduction of SBBC (green line) and estimated cumulative numbers of additional lives saved after introduction of SBBC for (**f**) Manyara, (**g**) Tabora, (**h**) Geita, (**i**) Shinyanga, and (**j**) Mwanza regions.

**Figure 7 children-10-00255-f007:**
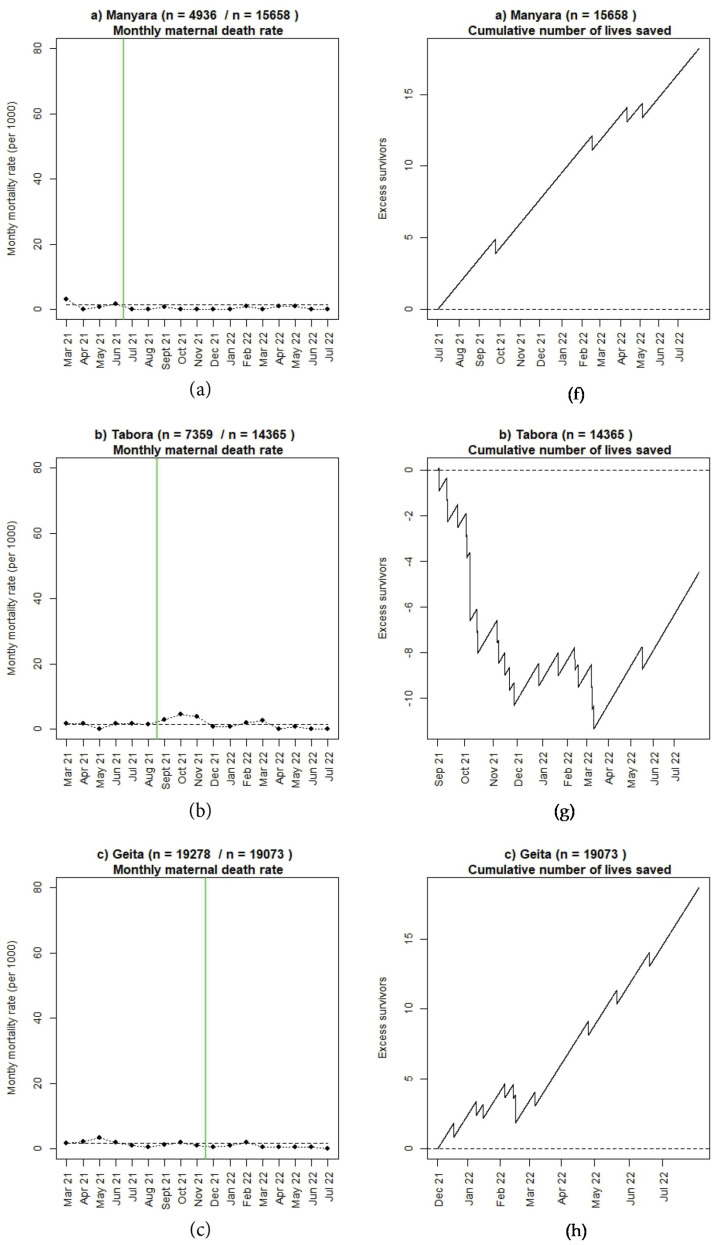
(**a**–**e**): Monthly maternal mortality rate before and after introduction of SBBC (green line) and estimated cumulative numbers of additional lives saved after introduction of SBBC for (**f**) Manyara, (**g**) Tabora, (**h**) Geita, (**i**) Shinyanga, and (**j**) Mwanza regions.

**Table 1 children-10-00255-t001:** Baseline key performance indicators, maternal and newborn characteristics, and outcomes across the 5 regions.

	Manyara	Tabora	Geita	Shinyanga	Mwanza
Time period	01.03.21– 30.06.21	01.03.21– 31.08.21	01.03.21– 30.11.21	01.03.21– 31.01.22	01.03.21– 31.02.22
Months of baseline	4	6	9	11	12
Deliveries (*n*)	4 936	7 359	19 278	15 544	20 573
Referred in labour from other facility	2.1 (104)	12.7 (935)	0.8 (162)	6.8 (1053)	1.8 (367)
Maternal age (years)	24 (20,30)	24 (20,30)	24 (20,30)	24 (20,30)	25 (21,30)
Parity (*n*)	1 (0,3)	2 (0,4)	2 (1,4)	1 (0,3)	1 (0,3)
Newborn birthweight (grams)	3100 (2900,3500)	3100 (2800,3500)	3200 (2900,3500)	3200 (2900,3500)	3000 (2700,3300)
**Fetal heart rate on admission**					
Abnormal (<110/>160 bpm)	0.9 (46)	0.9 (64)	0.2 (30)	0.3 (46)	0.2 (49)
Not detectable	1.2 (59)	3.2 (238)	1.3 (245)	1.4 (220)	1.6 (332)
Not recorded	9.9 (488)	9.2 (678)	16.2 (3113)	10.2 (1590)	19.3 (3957)
**Fetal heart rate during labour**					
Abnormal (<110/>160 bpm)	0.5 (25)	0.9 (64)	0.1 (28)	0.1 (21)	0.4 (71)
Not detectable	0.9 (44)	2.4 (178)	1.0 (187)	1.2 (186)	1.0 (201)
Not measured/recorded	34.9 (1726)	27.1 (1991)	25.9 (4993)	13.9 (2155)	26.5 (5461)
Moyo used	7.1 (348)	1.9 (138)	1.0 (201)	0 (0)	0 (6)
**Newborn Resuscitation**					
Stimulation/suction	13.7 (677)	22.0 (1625)	9.1 (1760)	10.8 (1673)	19.9 (4096)
Bag-mask ventilation	4.2 (204)	9.0 (671)	2.5 (490)	4.3 (675)	4.7 (972)
NeoBeat used	0.7 (37)	0.5 (37)	0.2 (38)	0 (0)	0.2 (49)
**Perinatal outcomes**					
Newborn deaths (24-h)	0.9 (46)	0.9 (63)	0.8 (151)	0.3 (40)	0.4 (81)
Fresh stillbirths	0.6 (30)	1.8 (132)	0.7 (141)	0.7 (114)	0.4 (83)
Maternal deaths	0.1 (7)	0.1 (10)	0.2 (30)	0.3 (43)	0.4 (75)
Maternal near miss	0.1 (4)	0.1 (9)	0.1 (13)	0.1 (18)	0.2 (34)

Values are presented as median (quartiles) or percent (numbers). All variables are significantly different (*p* < 0.0001) across regions. bpm = beats per minute.

**Table 2 children-10-00255-t002:** After initiation of SBBC key performance indicators, maternal and newborn characteristics, and outcomes across the 5 regions.

	Manyara	Tabora	Geita	Shinyanga	Mwanza
Time period	01.07.21– 31.07.22	01.09.21– 31.07.22	01.12.21– 31.07.22	01.02.22– 31.07.22	01.03.22– 31.07.22
Months of implementation	13	11	8	6	5
Deliveries (*n*)	15 658	14 365	19 073	8 453	13 118
Referred in labour from other facility	1.8 (277)	12.9 (1859)	1.0 (195)	4.5 (384)	1.6 (211)
Maternal age (years)	25 (21,30)	24 (20,30)	24 (20,30)	25 (20,30)	25 (21,30)
Parity (*n*)	1 (0,3)	2 (0,4)	2 (0,4)	1 (0,3)	1 (0,3)
Newborn birthweight (grams)	3200 (2900,3500)	3200 (2900,3500)	3200 (2800,3500)	3200 (2900,3500)	3000 (2700,3400)
**Fetal heart rate on admission**					
Abnormal (<110/>160 bpm)	1.8 (285)	1.6 (223)	0.3 (64)	0.5 (42)	0.7 (94)
Not detectable	1.0 (152)	3.0 (436)	1.5 (287)	1.6 (137)	1.6 (210)
Not recorded	5.7 (900)	8.7 (1260)	19.2 (3653)	17.8 (1502)	10.9 (1432)
**Fetal heart rate during labour**					
Abnormal (<110/>160 bpm)	1.0 (154)	1.1 (152)	0.2 (44)	0.4 (32)	0.5 (70)
Not detectable	0.8 (131)	2.2 (320)	1.3 (252)	1.7 (144)	1.2 (155)
Not recorded	23.1 (3615)	15.9 (2283)	27.6 (5275)	11.4 (963)	12.3 (1617)
Moyo used	65.7 (10,280)	35.5 (5094)	20.0 (3814)	44.2 (3732)	35.9 (4712)
**Newborn Resuscitation**					
Stimulation/suction	12.9 (2022)	10.5 (1506)	6.6 (1268)	8.8 (744)	12.8 (1678)
Bag-mask ventilation	3.9 (622)	6.0 (861)	3.6 (687)	5.0 (426)	4.7 (621)
NeoBeat used	9.6 (1498)	4.6 (683)	2.9 (555)	5.6 (471)	4.2 (556)
**Perinatal outcomes**					
Newborn deaths (24-h)	0.3 (52)	0.9 (136)	0.4 (69)	0.2 (13)	0.3 (36)
Fresh stillbirths	0.6 (93)	1.9 (276)	0.9 (165)	0.8 (66)	0.6 (84)
Maternal deaths	0 (4)	0.2 (24)	0.1 (11)	0.1 (5)	0.1 (7)
Maternal near miss	0.1 (18)	0.6 (80)	0.1 (14)	0.3 (28)	0.3 (37)

Values are presented as median (quartiles) or percent (numbers). All variables are significantly different (*p* < 0.0001) across regions. bpm = beats per minute.

**Table 3 children-10-00255-t003:** Source of admission, fetal heart rate on admission, use of Moyo, NeoBeat, and bag-mask ventilation among 24-h alive newborns, those who died within 24 h, and fresh stillborns during baseline and after the start of SBBC in Manyara and Tabora regions.

Manyara	Baseline *n* = 4936			After start SBBC *n* = 15,658		
	Alive *n* = 4860	Dead *n* = 46	FSB *n* = 30	Alive *n* = 15,513	Dead *n* = 52	FSB *n* = 93
**Source of admission**						
From antenatal care/maternity ward	1.4 (69)	2.2 (1)	3.3 (1)	5.2 (811)	9.6 (5)	4.3 (4)
From home	82.0 (3986)	73.9 (34)	73.3 (22)	79.4 (12,318)	75.0 (39)	76.3 (71)
Referred from other facility	2.0 (99)	8.7 (4)	3.3 (1)	1.7 (266)	0 (0)	11.8 (11)
Not recorded	14.5 (706)	15.2 (7)	20.0 (6)	13.7 (2118)	15.4 (8)	7.5 (7)
**Fetal heart rate on admission**						
Abnormal (<110/>160 bpm)	0.9 (45)	2.2 (1)	0 (0)	1.7 (268)	9.6 (5)	12.9 (12)
Not detectable	0.7 (35)	21.7 (10)	46.7 (14)	0.8 (121)	3.8 (2)	31.2 (29)
Not measured and/or recorded	9.8 (476)	19.6 (9)	10.0 (3)	5.7 (883)	11.5 (6)	11.9 (11)
**Use of new devices**						
Moyo	7.1 (346)	2.2 (1)	3.3 (1)	65.7 (10,193)	63.5 (33)	58.1 (54)
NeoBeat	0.7 (35)	4.3 (2)	0 (0)	9.4 (1456)	69.2 (36)	6.5 (6)
Bag-mask ventilation	3.8 (185)	34.8 (16)	10.0 (3)	3.7 (575)	78.8 (41)	6.5 (6)
**Tabora**	**Baseline *n* = 7359**			**After start SBBC *n* = 14,365**		
	Alive *n* = 7164	Dead *n* = 63	FSB *n* = 132	Alive *n* = 13,953	Dead *n* = 136	FSB *n* = 276
**Source of admission**						
From antenatal care/maternity ward	5.1 (362)	1.6 (1)	3.0 (4)	1.0 (139)	0.7 (1)	0.7 (1)
From home	82.2 (5890)	74.6 (47)	55.3 (73)	86.1 (12,015)	80.1 (109)	57.6 (159)
Referred from other facility	12.1 (867)	23.8 (15)	40.2 (53)	12.3 (1721)	17.6 (24)	41.3 (114)
Not recorded	0.6 (45)	0 (0)	1.5 (2)	0.6 (78)	1.5 (2)	0.4 (1)
**Fetal heart rate on admission**						
Abnormal (<110/>160 bpm)	0.9 (62)	1.6 (1)	0.8 (1)	1.4 (196)	5.9 (8)	6.9 (19)
Not detectable	2.2 (159)	19.0 (12)	50.8 (67)	2.2 (305)	6.6 (9)	44.2 (122)
Not measured and/or recorded	8.9 (634)	14.3 (9)	9.9 (13)	8.7 (1218)	10.3 (14)	10.1 (28)
**Use of new devices**						
Moyo	1.9 (135)	3.2 (2)	0.8 (1)	35.4 (4946)	27.2 (37)	40.2 (111)
NeoBeat	0.5 (34)	3.2 (2)	0.8 (1)	4.1 (568)	72.1 (98)	6.2 (17)
Bag-mask ventilation	8.6 (613)	63.5 (40)	12.9 (17)	5.2 (721)	83.8 (114)	9.4 (26)

Values are presented as percent (numbers). SBBC = Safer Births Bundle of Care, FSB = Fresh Stillborn, bpm = beats per minute.

## Data Availability

Data can be requested from Haydom Lutheran Hospital, through the project managers Paschal Mode and Benjamin Kamala.

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
