# Peer review of "“Safer Births Bundle of Care” Implementation and Perinatal Impact at 30 Hospitals in Tanzania—Halfway Evaluation"

_children, 2023, doi:10.3390/children10020255_

Round 1

Reviewer 1 Report

Which Type of randomization did you use?; (such as (1) simple, (2) block, (3) stratified and (4) unequal randomization or some other methods. please explain more.

Author Response

Thank you so much for reviewing our manuscript. 

Comment 1

Are all the cited references relevant to the research?

Response 1

Thank you for raising this concern. We thought it would be helpful for the readers to easy see the evidence-base for SBBC. However, we have gone through critically and have removed some of the references to previous studies that have been conducted to test different parts of the SBBC package.  

Comment 2

English language and style are fine/minor spell check required

Response 2

Thank you for notifying. English language has been corrected by a native English-speaking colleague.

Comment 3

Which Type of randomization did you use?; (such as (1) simple, (2) block, (3) stratified and (4) unequal randomization or some other methods. please explain more.

Response 3

We are a bit unsure about your question regarding type of randomisation. It is a step-wedge cluster randomisation, meaning that the 5 different wedges (regions) with its 6 clusters (=facilities) starts at different timepoints. Thus, it is a cluster randomisation type, as described in the text. The order of the regions (wedges) was drawn using simple randomisation, and this information has been added to the text; line 96.

Reviewer 2 Report

Very cost-effective intervention. well executed. 

Author Response

Thank you so much for reviewing our manuscript. 

Comment

Moderate English changes required

Response

Thank you for notifying. English language has been corrected by a native English-speaking colleague.

Reviewer 3 Report

this article entitled “Safer Births Bundle of Care” implementation and perinatal 2 impact at 30 hospitals in Tanzania - halfway evaluation "is very interesting and valuable topic, however ,there are some concern need be addressed:

1 Fig s1 should be Fig 1, in stand,Fig 1 should be Fig s1, all other figs need revise like this.

2  in the current fig s2 ,photograph should be acquire permission from the source, please explain this or note it.

3 in statistic section "Data are analysed using R-4.2.1 software ",please mark the company and location.also in this section ,P-value need define in detail as they are in the text.

4 in table 1 and 2 ,I suggest you can analyze statistic significance result in table.

5 in all tables ,you need note clearly regarding value ,eg Maternal age (years) ,25 (21,30) please mark in detail ,what 's did they stand for in the first table show?

Author Response

Thank you so much for reviewing our manuscript.

Comment 1

Moderate English changes required

Response 1

Thank you for notifying. English language has been corrected by a native English-speaking colleague.

Comment 2

The reviewer has crossed “can be improved” for:

Does the introduction provide sufficient background and include all relevant references?

Are all the cited references relevant to the research?

Is the research design appropriate?

Are the methods adequately described?

Are the results clearly presented?

Response 2

We have tried to clarify all these issues throughout the manuscript, and we have marked all the changes in the revised version. Regarding references, we thought it would be helpful for the readers to easy see the evidence-base for SBBC. However, we have gone through critically and have removed some of the references to previous Safer Births studies that have been conducted to test different parts of the SBBC package.  

We are a bit unsure about your question regarding research design. It is a step-wedge cluster randomised implementation trial, meaning that the 5 different wedges (regions) with its 6 clusters (=facilities) starts implementation of the bundle at different timepoints. 

Comment 3

The reviewer has crossed “must be improved” for:

Are the conclusions supported by the results?

Response 3

As this manuscript is a halfway evaluation of an ongoing step-wedge cluster randomised implementation project, we cannot conduct any direct analyses/tests, at the current stage, on the primary outcomes of the trial (i.e. newborn mortality, fresh stillborn, and maternal mortality). The 3 main hypotheses are described in the text. The trial power and sample size calculations only allow for one final testing of the end-results. However, setting up the trial, we planned to do this halfway evaluation to describe differences across the 5 regions (which is presented in the tables with statistical tests) and to describe trends in adoption of new devices and perinatal mortality using statistical process control methods (which is presented in figures 3-7). Such statistical process control methos provide great insights into how indicators are changing over time (improving or not). Combined with the key performance variables (presented in the tables) it is possible to better understand how/what to adjust/strengthen as we move forward. This is part of our trial design. Based on the absolute numbers of deaths (as presented in the tables) and as shown in the mortality trend figures 5-7, we see that the newborn mortality is approximately halved, and that the maternal mortality is reduced by 10-20%. Based on the implementation design of the trial and the analyses done, we think it is appropriate to say that “this halfway evaluation of SBBC indicates steady trends toward a 50% reduction in 24-hour newborn mortality and a 10-20% reduction in maternal mortality in four of five regions.” And “Despite increased survival rates in line with our hypotheses, these preliminary data indicate that there is a remaining impact potential if uptake of the bundle is further strengthened over time”.

Comment 4

Fig s1 should be Fig 1, in stand,Fig 1 should be Fig s1, all other figs need revise like this.

Response 4

Thank you for notifying. This has been correct throughout the manuscript.

Comment 5  

in the current fig s2 ,photograph should be acquire permission from the source, please explain this or note it.

Response 5

Thank you for notifying. The required information has been added as a note below the figure.

Comment 6

in statistic section "Data are analysed using R-4.2.1 software ",please mark the company and location.

Response 6

As “R” is an open-source software, not a commercial product, the developers of the free “R” software want to be credited by the reference, as we have done. (26) R Core Team (2022). R: A language and environment for statistical computing. R Foundation for Statistical Computing, Vienna, Austria. URL https://www.R-project.org/.

However, we have added to the text that “R” is an “open-source” software.

Comment 7

also in this section ,P-value need define in detail as they are in the text.

Response 7

Thank you for notifying. We have added; “P<0.05 is used as level of significance.”

Comment 8

in table 1 and 2 ,I suggest you can analyze statistic significance result in table.

Response 8

Thank you for this valuable comment. We have analysed differences across the 5 regions. All variables are significantly different with p<0.0001, and this is stated below the tables. If this important information is missed, it can be included as a last (7th) column in the tables 1 and 2, stating p<0.0001 for all variables?

But, this will enlarge the tables, and I found it difficult to fit into the page.

Comments 9

in all tables ,you need note clearly regarding value ,eg Maternal age (years) ,25 (21,30) please mark in detail ,what 's did they stand for in the first table show?

Response 9

Thank you for this important input. We have tried to update the tables as requested. It is stated below the tables that “Values are presented as median (quartiles) or percent (numbers)”

Reviewer 4 Report

Dear authors,

Neonatal death is a very important problem of contemporary world and the subject deserves to be studied and discussed. Every effort must be made to reduce perinatal death rates in underdeveloped and developing countries.

Regarding the manuscript, I think it is well structured, well written and deserves to be published. We are waiting for the final results and hoping for improvement in neonatal care in Tanzania and underdeveloped countries. 

Author Response

Thank you so much for reviewing our manuscript and for the nice words.

Round 2

Reviewer 3 Report

accept